# Yaws elimination in Ecuador: Findings of a serological survey of children in Esmeraldas province to evaluate interruption of transmission

**Philip J. Cooper** [1,2]*, **Mariella Anselmi**[3], **Cintia Caicedo**[3], **Andrea Lopez**[1], **Yosselin Vicuña**[4], **Jaen Cagua Ordoñez**[5,6], **Julio Rivera Bonilla**[5], **Alejandro Rodriguez**[1], **Aida Soto**[7], **Angel Guevara**[4]

1 Escuela de Medicina, Universidad Internacional del Ecuador, Quito, Ecuador, 2 Insititute of Infection and Immunity, St George's University of London, London, United Kingdom, 3 Centro de Epidemiologia Comunitaria y Medicina Tropical (CECOMET), Esmeraldas, Ecuador, 4 Instituto de Biomedicina, Carrera de Medicina, Universidad Central, Quito, Ecuador, 5 Dirección Nacional de Estrategias de Prevención y Control, Ministerio de Salud Pública, Quito, Ecuador, 6 Instituto Ecuatoriano de Seguridad Social, Quito, Ecuador, 7 Pan American Health Organization, Quito, Ecuador

* pcooper@sgul.ac.uk

**Data Availability Statement:** All relevant data are within the manuscript and its Supporting Information files.

## Abstract

### Background

The WHO roadmap for neglected tropical diseases includes yaws eradication requiring certification of elimination of transmission in all endemic and formerly endemic countries worldwide. A community-based programme for yaws control was considered to have achieved elimination of the infection in the endemic focus in Ecuador after 1993. We did a serosurvey of children in this focus to provide evidence for interruption of transmission.

### Methods

Survey of serum samples collected from children aged 2 to 15 years living in the formerly endemic and in geographically contiguous areas. A convenience sample of sera collected between 2005 were 2017 from non-yaws studies, were analyzed using immunochromatic rapid tests to screen (OnSite Syphilis Ab Combo Rapid Test) for *Treponema pallidum*-specific antibodies and confirm (DPP Syphilis Screen and Confirm) seroreactivity based on the presence antibodies to treponemal and non-treponemal antigens.

### Results

Seroreactivity was confirmed in 6 (0.14%, 95% CI 0.06–0.30) of 4,432 sera analyzed and was similar in formerly endemic (0.11%, (95% CI 0.01–0.75) and non-endemic (0.14%, 95% CI 0.06–0.34) communities. All seroreactors were of Afro-Ecuadorian ethnicity and most were male (4/6) and aged 10 or more years (5/6), the latter possibly indicating venereal syphilis. Only 1 seroreactor lived in a community in the Rio Santiago, that was formerly hyperendemic for yaws.

**Funding:** This work was supported by Wellcome Trust (https://wellcome.org) grants 074679/Z/04/Z (PJC) and 088862/Z/09/Z (PJC). Rapid diagnostic tests were purchased by the Pan American Health Organization and supplied through the Ecuadorian Ministry of Public Health. The funders had no role in study design, data collection and analysis, decision to publish, or preparation of the manuscript.

**Competing interests:** The authors have declared that no competing interests exist.

## Conclusion

We observed very low levels of treponemal transmission in both formerly endemic and non-endemic communities which might be indicative of congenital or venereal syphilis and, if yaws, would likely be insufficient to maintain transmission of this endemic childhood infection. Additional surveys of children aged 1 to 5 years are planned in Rio Santiago communities to exclude yaws transmission.

### Author summary

Yaws, caused by infection with the spirochete, *Treponema pallidum pertenue*, causes a chronic debilitating condition of skin, cartilage, and bone, and is transmitted during childhood through skin-to-skin contact. Yaws has been targeted for eradication as part of the WHO roadmap for control of neglected tropical diseases, requiring certification of elimination in all endemic and formerly endemic regions. Yaws in Ecuador has been restricted to a geographically isolated focus in a rainforest region of Esmeraldas Province in northern coastal Ecuador. Following a strategy of repeated 5-yearly clinical and serological surveys with mass-treatment and surveillance between surveys, yaws was assumed to have been eliminated by 1998. To provide the evidence base to certify the elimination of transmission in Ecuador, this study presents an analysis of stored sera collected from 4,432 children between 2005 and 2017 from formerly endemic and non-endemic communities. Screening and confirmation of seroreactors was done using two validated rapid tests for *T. pallidum*. Seroreactivity was observed in 6 samples (0.14%) and was similar in formerly endemic (0.11%) and non-endemic (0.14%) communities, possibly explained by background rates of congenital or venereal syphilis. Only 1 active infection was detected in formerly endemic communities. To our knowledge, this is the first study of yaws from the Americas to evaluate the elimination of transmission. Our data indicate that active yaws transmission is unlikely to be occurring in formerly endemic communities. Additional surveys of young children may be required to confirm interruption of transmission.

## Introduction

Yaws, caused by the spirochete *Treponema pallidum* subspecies *pertenue*, is a chronic debilitating disease of skin, cartilage, and bone, transmitted during childhood through skin-to-skin contact. Yaws was targeted for eradication in Latin America and other world regions in the 1950s but remained endemic in a geographically isolated region of Esmeraldas Province in North-Western coastal Ecuador [1]. During a yaws survey of this region in 1988, active cases were identified in a geographically restricted area in a single District [2]. Mass treatment was given with benzathine penicillin according to WHO recommendations [3]. This was accompanied by a control programme of clinical and serological resurveys at 5-year intervals with mass treatments for communities with active cases. Surveillance was done by community health workers during inter-survey periods with treatment of cases and contacts [4]. No active cases were detected after the 1993 survey and active surveillance was maintained to 1998 when a repeat survey showed no active cases and negligible low-titer seropositivity, indicating likely interruption of yaws transmission [5].

There has been renewed interest in yaws eradication since 2012 when WHO launched a roadmap for the control of neglected tropical diseases [6]. A clinical trial showing high

effectiveness of oral azithromycin in treating clinical yaws [7] led to the Morges strategy for the eradication of yaws. This strategy was based on mass treatments with azithromycin of communities with active cases (or total community treatment) followed by resurveys with treatment of cases and contacts [8].

Worldwide eradication will require certification of elimination of transmission in all currently endemic and formerly endemic countries including Ecuador. A meeting of experts (representing universities, a non-governmental organization [NGO], the Ministry of Public Health [MSP] and the Pan American Health Organization) was held in Quito, Ecuador, on 30 May 2018, to review gaps in information to support the interruption of yaws transmission. At the meeting it was decided to use validated rapid diagnostic tests for *T. pallidum* following WHO guidelines [9] to screen stored sera to provide evidence for interruption of transmission. Sera collected after 1998 from children living in the formerly endemic yaws focus, and in geographically close areas in three Districts in Esmeraldas province, were therefore screened.

## Methods

### Ethics statement

The protocols of each of the three surveys used for the present analysis were approved by local ethics committees in Ecuador as described [10,12,13]. The protocol for this retrospective analysis of stored samples was approved by the Ethics Committee of the Universidad Central del Ecuador (Reference 001-SEISH-UCE-20). Informed written consent was obtained from the child's parent or guardian for participation in each of the three studies. Informed consent in all three studies included storage and analysis of blood samples for future studies.

### Study populations

Samples were collected between 2005 and 2017 as part of two observational studies of allergy in children in the Districts of Eloy Alfaro, San Lorenzo, and Quininde in Esmeraldas Province: the first study was a cross-sectional survey of school age children aged 6 to 15 years (predominantly of Afro-Ecuadorian ethnicity in the Districts of Eloy Alfaro and San Lorenzo including formerly endemic and non-endemic communities) using updated censuses of each community with blood sampling between 2005 and 2009 [10] while the second was a population-based birth cohort (predominantly of mestizo ethnicity in the District of Quininde in geographically close but non-endemic communities) in which serum samples collected between 2014 and 2017 were analysed at 8 to 9 years [11,12]. Sera from a survey of indigenous Awa children aged 2 to 15 years in the District of San Lorenzo (of unknown endemicity for yaws) collected in 2014 and 2015 were analyzed also [13]. Frozen sera from these studies were selected based on living in a previously endemic community for yaws [2], location of yaws cases from historical records of the Ecuadorian Ministry of Public Health, and geographical proximity to these locations.

### Serological analysis

Stored sera were tested for the presence of antibodies to treponemal and non-treponemal antigens using two rapid immunochromatic lateral flow diagnostic tests as recommended by WHO [9]. Screening of all samples was done using OnSite Syphilis Ab Combo Rapid Test (CTK Biotech, San Diego, CA), that detects *T. pallidum*-specific antibodies, and confirmation of positives was done using DPP Syphilis Screen and Confirm (ChemBio Diagnostics, Hauppauge, NY) that detects antibodies to *T. pallidum* and non-treponemal antigens, following the manufacturers' instructions. A positive confirmatory test showed a positive test for specific

antibodies to both *T. pallidum* and non-treponemal antigens (referred to here as seroreactors or active infections). Individuals whose sera had *T. pallidum*-specific antibodies only were interpreted as having past or treated infections. Rapid tests were read by eye in a well-lit area with natural light. All positive or invalid screening and confirmatory tests were repeated twice to confirm results. The rapid tests were purchased by the Pan American Health Organization (PAHO) and supplied through the Ministry of Public Health and a training course was provided on the use of the rapid tests through PAHO.

## Statistical analysis

This study used a convenience sample based on availability of archived samples collected from previous non-yaws surveys done in the region and 4,500 rapid screening tests were provided through the Ecuadorian MPH. Lists of available sera samples from children aged 2 to 15 years living in communities in the formerly endemic or contiguous areas was drawn up, anonymized, and random samples were selected using Stata (version 11). Seroreactive individuals were defined using the results of the confirmatory test as having specific antibodies to both *T. pallidum* and non-treponemal antigens. All other individuals were defined as seronegative. Seroreactivity in endemic and non-endemic samples was compared with a population estimate of 0.37% [14] using the binomial probability test with a two-tailed P value. A separate analysis looked at individuals with either evidence of active (i.e seroreactors) or past/treated infections (i.e presence of antibodies only to *T. pallidum* antigens).

## Results

A total of 4,432 serum samples were screened using the 'OnSite Syphilis Ab Combo Rapid Test': 53 samples (1.2%) gave a positive result (S1 Table). All were positive on repeat testing. Confirmation of positivity on these samples was done using 'DPP Syphilis Screen and Confirm assay' giving the following results: 33 negatives; 2 false positives (positive only for antibodies to non-trepenomal antigens); 12 positives just for antibodies to *T. pallidum* antigen (representing previously treated or cured *T. pallidum* infections); and 6 were seroreactors and positive for antibodies to *T. pallidum* and non-treponemal antigens (representing active *T. pallidum* infections). Findings of seroreactivity in communities from formerly endemic and non-endemic communities in Esmeraldas Province with age and sex distributions are shown in Table 1, and the locations of these communities are shown in Fig 1. Sera samples from only 3 of the formerly endemic communities were not available for this analysis (Fig 1).

Seroreactivity was observed in 0.14% (95% CI 0.06–0.30) of all children tested: only 1 (0.11%, 95% CI 0.01–0.75) seroreactive individual (detected in 2005 in Selva Alegre, Rio Santiago) was detected in formerly endemic compared to 5 (0.14%, 95% CI 0.06–0.34) in non-endemic communities (Table 1). Compared to a population estimate of *T. pallidum* seropositivity of 0.37% derived from women of reproductive age from coastal Ecuador [14], seroreactivity was significantly less in non-endemic (P = 0.024) but not endemic (P = 0.278) communities. All sero-reactors were of Afro-Ecuadorian ethnicity and most were male (4/6) and aged 10 or more years (5/6) (Table 2). Most sera were collected between 2005 and 2009 (83.1% of samples): sampling in formerly endemic areas was done between 2005 and 2009 while sampling in non-endemic areas was done between 2005 and 2017 (S2 Table). The last detected seroreactive was from a sample collected in 2007.

Analysis of any positivity to *T. pallidum* antigens including treated/cured infections could provide useful information about the history of *T. pallidum* infection in a population, given that these antibodies can persist for many years. The positivity rate in the whole sample was 0.41% (95% CI 0.26–0.64): 0.53% (95% CI 0.22–1.26) in formerly endemic and 0.37% (95% CI

**Table 1. Seroreactivity to *T. pallidum* using the confirmatory rapid test (DPP Syphilis Screen and Confirm) in 4,432 schoolchildren aged 2 to 15 years living in formerly endemic\* and non-endemic communities for yaws.**

| Communities screened | Sample | Sex (Male/female) % | Median age (range) yrs. | Seroreactivity (%) N | % [95%CI] |
|---|---|---|---|---|---|
| All areas | 4,432 | 51/49 | 10 (2–15) | 6 | 0.14 [0.06–0.30] |
| Formerly endemic region | | | | | |
| All communities | 947 | 54/46 | 11 (5–15) | 1 | 0.11 [0.01–0.75] |
| Rio Santiago | | | | | |
| Playa de Oro | 40 | 58/42 | 10 (7–15) | 0 | 0 |
| Angostura | 14 | 64/36 | 10 (5–14) | 0 | 0 |
| Playa Tigre/Playa Nueva/Zapote | 45 | 51/49 | 9 (6–15) | 0 | 0 |
| Palma Real/Guayabal | 37 | 65/35 | 11 (7–15) | 0 | 0 |
| Chanuzal/Pailon/Picadero | 31 | 45/55 | 11 (7–15) | 0 | 0 |
| Selva Alegre | 122 | 51/49 | 11 (7–15) | 1 | 0.82 [0.11–5.59] |
| Timbire/El Porvenir | 121 | 52/48 | 10 (7–15) | 0 | 0 |
| Las Antonias | 27 | 56/44 | 9 (7–15) | 0 | 0 |
| Rocafuerte | 31 | 42/58 | 10 (6–13) | 0 | 0 |
| Rio Cayapas | | | | | |
| San Miguel | 33 | 49/51 | 11 (7–15) | 0 | 0 |
| Mafua | 15 | 53/47 | 9 (7–15) | 0 | 0 |
| Zapallo Grande | 78 | 55/45 | 12 (7–15) | 0 | 0 |
| Telembi | 74 | 57/43 | 12 (7–15) | 0 | 0 |
| Rio Zapallito | | | | | |
| Boca de Zapallito | 45 | 60/40 | 10 (7–15) | 0 | 0 |
| Rio Onzole | | | | | |
| Colon | 131 | 52/48 | 11 (7–15) | 0 | 0 |
| Santo Domingo | 103 | 61/39 | 11 (7–15) | 0 | 0 |
| Contiguous regions | | | | | |
| All communities | 3,485 | 50/50 | 10 (2–15) | 5 | 0.14 [0.06–0.34] |
| District of Eloy Alfaro | | | | | |
| Rio Cayapas | | | | | |
| 8 communities | 304 | 54/46 | 10 (6–15) | 0 | 0 |
| Rio Santiago | | | | | |
| 3 communities | 340 | 53/47 | 11 (6–15) | 1 | 0.29 [0.04–2.06] |
| Rio Onzole | | | | | |
| 3 communities | 155 | 48/52 | 10 (6–15) | 0 | 0 |
| Others | | | | | |
| 19 communities | 1,059 | 51/49 | 11 (6–15) | 4 | 0.38 [0.14–1.00] |
| District of San Lorenzo | | | | | |
| 16 communities | 1,154 | 47/53 | 10 (2–15) | 0 | 0 |
| District of Quininde | | | | | |
| 10 communities | 473 | 49/51 | 8 (8–13) | 0 | 0 |

\*Endemicity defined by presence of active yaws lesions in 1988 survey [2].

0.22–0.64) in non-endemic communities (S3 Table). There was some evidence of an increase in antibody positivity to *T. pallidum* antigens by age (S1A Fig) that was clearer in non-endemic (S1C Fig) than endemic communities (S1B Fig). No positives were detected among the youngest 132 children aged 2 to 6 years. Although most communities in the formerly endemic area had no evidence of positivity, 4 did with positivity rates ranging 0.76% to 2.70%. The upper level of the 95% confidence intervals indicates positivity of up to 16.85%. Among non-endemic communities geographically contiguous with formerly endemic communities in Rio Santiago and Rio Onzole, positivity varied 0.17% to 1.47%. There were no positives among a sample of Awa Amerindians in the District of San Lorenzo. Characteristics of all individuals with positive tests for antibodies to *T. pallidum* antigens are shown in S4 Table. All positives were of Afro-Ecuadorian ethnicity, 12/18 were aged greater than 10 years, 13/18 were boys, 5/18 were from formerly endemic communities, and 16/18 were from communities in the District of Eloy Alfaro. Five communities on the Rio Santiago accounted for half the positives.

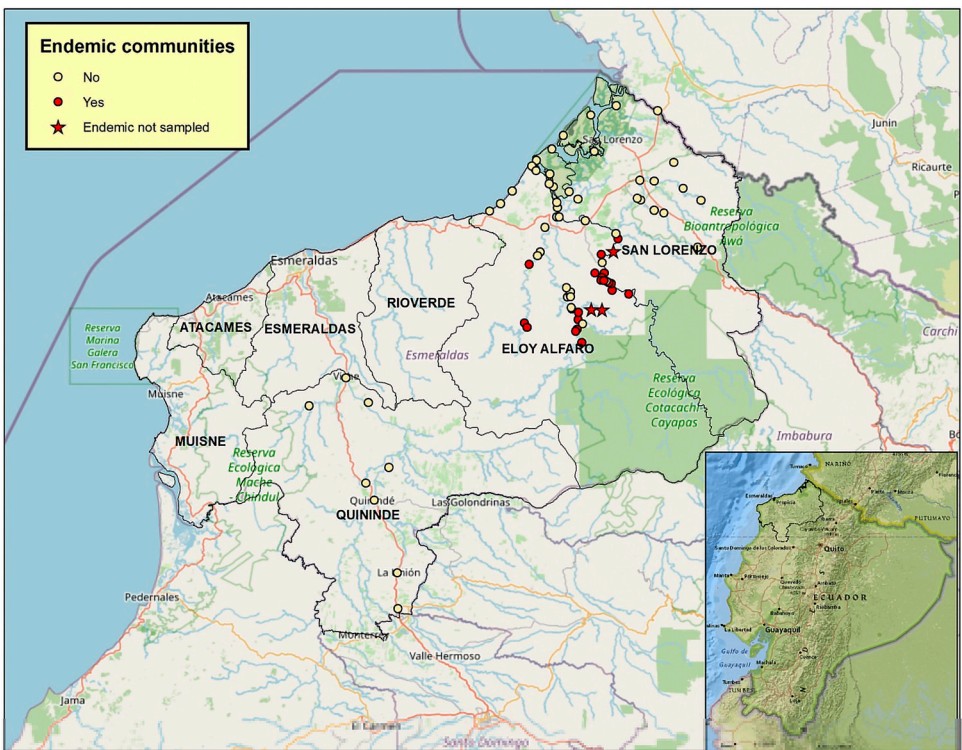

**Fig 1. Map of formerly endemic and non-endemic sample communities in Esmeraldas Province. Shown are locations of formerly endemic (red circles) and non-endemic (yellow circles) communities, and formerly endemic communities not sampled (red stars).** The map was built using ArcGIS version 10.2.2 (ESRI, California, USA).

## Discussion

Yaws is a chronic debilitating neglected tropical disease that has been targeted for worldwide eradication by 2030 [15]. Yaws was formerly endemic in a small geographically restricted and isolated focus in Esmeraldas Province in north-western coastal Ecuador [2]. A control programme implemented from 1988 was assumed to have eliminated the infection [5] although elimination has not yet been formally certified. We present here data from a serosurvey of a sample of 4,432 schoolchildren aged 2 to 15 years to detect possible evidence of yaws transmission in formerly endemic and non-endemic communities in this region of Esmeraldas Province. The survey, done using two rapid diagnostics tests recommended by WHO for the surveillance of yaws, showed low rates of seroreactivity that were similar in formerly endemic (0.11%) and non-endemic (0.14%) communities. All seroreactives were of Afro-Ecuadorian ethnicity and most were boys aged 10 or more years.

**Table 2. Characteristics of seroreactive individuals using the confirmatory rapid test (DPP Syphilis Screen and Confirm).**

| Age (years) | Ethnicity | Community | Region | Sex | Sample year | Formerly endemic |
|---|---|---|---|---|---|---|
| 10 | Afro-Ecuadorian | Canchimalero | Eloy Alfaro | Male | 2007 | No |
| 8 | Afro-Ecuadorian | Cuerval | Eloy Alfaro | Male | 2006 | No |
| 12 | Afro-Ecuadorian | Maldonado | Rio Santiago (Eloy Alfaro) | Female | 2005 | No |
| 15 | Afro-Ecuadorian | Rompido | Eloy Alfaro | Male | 2006 | No |
| 12 | Afro-Ecuadorian | Santa Rosa | Eloy Alfaro | Male | 2006 | No |
| 14 | Afro-Ecuadorian | Selva Alegre | Rio Santiago (Eloy Alfaro) | Female | 2005 | Yes |

The control programme for yaws in Esmeraldas Province was implemented in the late 1980s. A survey of villages with sporadic official reports of yaws cases in 1988 identified a prevalence of 16.5% with active yaws lesions and seropositivity of 96.3% [2]. The survey was accompanied by mass treatment with intramuscular benzathine penicillin according to level of endemicity following WHO recommendations [3]. Yaws was subsequently incorporated into the surveillance activities of a community health programme centred around a network of community health workers (CHWs) or 'health promoters'. Surveillance was based on continuous active case finding by CHWs in which confirmed cases and close contacts were treated with intramuscular benzathine penicillin. Clinical and serological surveys for yaws were repeated at 5-year intervals (in 1993 and 1998) by health teams (from Vicariato Apostolico de Esmeraldas or CECOMET in coordination with MSP) in previously affected communities [5]. Mass treatments were given to those communities still with active cases. The second survey in 1993 identified 16 active cases in only 5 communities representing 1.4% of those examined and seropositivity of 4.7% [5]. No active cases were detected after 1993 and a repeat survey in 1998 showed a seroprevalence of 3.5%: those with positive serology at this time were presumed to have had previously treated syphilis or yaws with low-level antibody titers of less than 1:8 [5]. Active surveillance for yaws after 1998 was continued through CHWs but no new active cases were confirmed, indicating presumed interruption of yaws transmission [5]. CHWs were key to the implementation and execution of this control programme. CHW training was based on essential techniques and instruments of community epidemiology [16]. Yaws control activities were integrated within other activities that included surveillance, management, and control of malaria, tuberculosis, onchocerciasis, chronic non-communicable diseases, and maternal and child health.

Serological tests cannot distinguish between syphilis and yaws. The confirmatory test has been shown to be highly specific with high positive and negative predictive values [17]. Clinically active and latent (indicated here by serological evidence of active infection) yaws represent the reservoir of infection required to sustain transmission. No clinical data were available to determine presence of possible yaws lesions. Serologic evidence of active infections (i.e. seroreactors) was present in only 6 of all 4,432 (0.14%) children tested and in only 1 of 947 (0.11%) children from the formerly endemic communities. All these samples were collected before 2008. An explanation for a positive confirmatory test could be explained by congenital or sexually transmitted syphilis, the latter in minors if sexually abused. Rates of active infections (based on presence of antibodies to both *T.pallidum* and non-treponemal antigens) with syphilis in women of reproductive age have been estimated at 0.37% in women from coastal Ecuador [14]. If no treatment is given during pregnancy, as might be expected in most of the survey communities, transmission rates to offspring can reach 100% [18]. There are no reliable epidemiological data on congenital syphilis for the study area. The observation that the majority of seropositives occurred among those aged greater than 10 years could favour venereal transmission of syphilis as the primary cause of positive tests. We examined rates of positivity for antibodies to *T. pallidum* antigens, that persist for many years even after cure, as an indicator of the history of treponemal infections in these communities. *T. pallidum pertenue* is highly sensitive to several antibiotics [19], the wide use of which for other indications, could have modified the seroepidemiology of the infection in this setting. A gradual increase in positivity was observed with greater age in non-endemic but not formerly endemic communities–such a pattern could indicate either low-level transmission of yaws or sporadic cases of venereal syphilis with risk increasing with age. However, the fact that no active clinical cases of yaws have been reported since 1993 despite continued surveillance argues strongly against active transmission of yaws, although more attenuated forms of disease [20] might be overlooked.

It is unclear how large an infectious reservoir in humans is required to maintain transmission of yaws. Yaws transmission likely requires a sufficiently large reservoir of individuals with latent yaws (with repeated infectious relapses) living in sufficiently close contact with susceptible individuals in suitable social and environmental conditions. Such conditions would be present in populations living in severe poverty with close physical contact and poor hygiene in humid tropical climates. Previous studies have suggested that a ratio of 6 latent to 1 active cases may be typical for endemic transmission [21]. However, our data indicate that no such reservoir presently exists and social and economic conditions have altered dramatically over the past 20 years in the formerly endemic area—including widespread access to clean water, sanitation, housing, education, and basic health services including antibiotics–to such a degree that transmission even if initiated would be unlikely to be sustained. It seems improbable that such small numbers of seroreactors with active infections (i.e. 6/4,432) would be sufficient to maintain active transmission. Sporadic cases could conceivably result from contact with non-human primates, that have been shown to harbour *T. pallidum pertenue*-like strains of spirochetes [22]. Such contacts are known to be very infrequent in this region.

The Ecuador control and elimination strategy was similar to that proposed subsequently by WHO in 2012 as the Morges strategy [8]. WHO criteria for certification of yaws elimination include absence of: 1) confirmed cases detected for 3 consecutive years through high coverage with active surveillance; and 2) transmission as measured by serosurveys with evidence of continuous negative serological tests for at least 3 consecutive years in samples of asymptomatic children aged 1 to 5 years [9,14] (i.e. those born since the implementation of elimination strategies). In this study, we evaluated seroreactivity in children aged 2 to 15 years from formerly endemic and geographically close communities–all seroreactive children had been born after the last reported case in the community where they lived [5]. Current WHO guidelines for certification of elimination provide no recommendations for how to deal with low rates of seroreactivity in settings where yaws transmission may have been eliminated but where congenital syphilis is present. This is complicated by the fact that populations that have suffered endemic yaws tend to be extremely poor, live in geographically isolated settings, and have limited access to health care. Data on rates of congenital and venereal syphilis are likely to be inadequate in most settings but screening for syphilis during pregnancy is widely implemented. We recommend that in the absence of local epidemiological data on rates of syphilis in pregnant women in such populations, that the latest national data is used to estimate background rates of congenital syphilis (or venereal syphilis if children up to 15 years are included in such surveys) to define the upper limits of serological thresholds of concern. Seroreactivity 'mapping' as done here could then be used to identify communities where rates are higher than might be expected and which might require additional epidemiological studies.

In conclusion, our data indicate evidence for the presence of very low levels of active treponemal infections, based on treponemal serology, among schoolchildren in a region of Ecuador where yaws was formerly endemic and where a successful control programme eliminated clinical yaws. Higher rates of active infections were observed in some communities in or contiguous with the formerly endemic area, but serological tests are unable to distinguish between yaws and venereal or congenital syphilis. It appears unlikely that such low rates of treponematosis, if truly *T. pallidum pertenue*, would be sufficient to sustain yaws transmission, thus indicating no evidence for active transmission of yaws in the communities where this study was done. Serosurveys of young children aged 1 to 5 years accompanied by detailed clinical and laboratory evaluation of seroreactive children, are now planned in communities in the Rio Santiago, to exclude the persistence of yaws transmission.

## Supporting information

**S1 Table. Results of screening tests for antibodies to *T. pallidum* antigens in 4,432 school-children aged 2 to 15 years living in formerly endemic and non-endemic communities for yaws using OnSite Syphilis Ab Combo.**
(DOCX)

**S2 Table. Timing of sera collection by calendar year and seropositivity stratified by endemicity for yaws.**
(DOCX)

**S3 Table. Serological results using the confirmatory rapid test (DPP Syphilis Screen and Confirm) in 4,432 schoolchildren aged 2 to 15 years living in formerly endemic* and non-endemic communities for yaws.** Results show serological reactivity to *T. pallidum* antigens (any positive) further stratified into presence (active infections) or absence (past infections) of non-treponemal antibodies.
(DOCX)

**S4 Table. Characteristics of all individuals with positivity for antibodies to *T. pallidum* antigens with and without antibodies to non-treponemal antigens using the confirmatory rapid test (DPP Syphilis Screen and Confirm).**
(DOCX)

**S1 Fig. Bar charts showing changes in any antibody positivity to *T. pallidum* antigens by age in 4,432 schoolchildren aged 2 to 15 years.** A–age-positivity in all children and stratified into children living in formerly endemic (B) and non-endemic (C) communities. Numbers in brackets represent sample numbers for each age.
(TIF)

**S1 Data. Raw data used for analyses.**
(XLSX)

## Acknowledgments

AS is a staff member of the Pan American Health Organization. The authors alone are responsible for the views expressed in this publication, and they do not necessarily represent the decisions or policies of the Pan American Health Organization.

## Author Contributions

**Conceptualization:** Philip J. Cooper, Mariella Anselmi, Aida Soto, Angel Guevara.

**Data curation:** Philip J. Cooper, Angel Guevara.

**Formal analysis:** Philip J. Cooper, Alejandro Rodriguez.

**Funding acquisition:** Philip J. Cooper, Mariella Anselmi, Aida Soto.

**Investigation:** Philip J. Cooper, Mariella Anselmi, Cintia Caicedo, Andrea Lopez, Yosselin Vicuña, Angel Guevara.

**Methodology:** Philip J. Cooper, Mariella Anselmi, Angel Guevara.

**Project administration:** Philip J. Cooper, Mariella Anselmi, Cintia Caicedo.

**Resources:** Philip J. Cooper, Mariella Anselmi, Jaen Cagua Ordoñez, Julio Rivera Bonilla, Angel Guevara.

**Supervision:** Philip J. Cooper, Mariella Anselmi, Angel Guevara.

**Visualization:** Alejandro Rodriguez.

**Writing – original draft:** Philip J. Cooper.

**Writing – review & editing:** Philip J. Cooper, Mariella Anselmi, Cintia Caicedo, Andrea Lopez, Yosselin Vicuña, Jaen Cagua Ordoñez, Julio Rivera Bonilla, Alejandro Rodriguez, Aida Soto, Angel Guevara.

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
