## [Decision Letter · Decision Letter 0]

15 Mar 2022

Dear Dr. Cooper,

Thank you very much for submitting your manuscript "Yaws elimination in Ecuador: findings of a serological survey of children in Esmeraldas Province to evaluate interruption of transmission" for consideration at PLOS Neglected Tropical Diseases. As with all papers reviewed by the journal, your manuscript was reviewed by members of the editorial board and by several independent reviewers. The reviewers appreciated the attention to an important topic. However, the reviewers raise a few concerns, including the lack of statistical analysis and the inconsistent use of definitions such as seropositivity. The reviewers also suggest that the manuscript be proofread more carefully. Overall, the reviews are generally positive and we encourage you to submit a modified manuscript according to the review recommendations. 

Sincerely,

Brian Stevenson, Ph.D.

Associate Editor

Melissa Caimano, Ph.D.

Deputy Editor

Reviewer's Responses to Questions

**Key Review Criteria Required for Acceptance?**

**Methods**

-Are the objectives of the study clearly articulated with a clear testable hypothesis stated?

-Is the study design appropriate to address the stated objectives?

-Is the population clearly described and appropriate for the hypothesis being tested?

-Is the sample size sufficient to ensure adequate power to address the hypothesis being tested?

-Were correct statistical analysis used to support conclusions?

-Are there concerns about ethical or regulatory requirements being met?

Reviewer #1: (No Response)

Reviewer #2: The methods do not note plasma samples collected, but plasma samples are specifically mentioned in the introduction and results. Please clarify (either add plasma to methods or delete plasma elsewhere).

Reviewer #3: The study objectives are clear with rationale explained. The study populations are described, but the studies where samples were collected should be cited. The study design is appropriate but no sample size calculation has been included to demonstrate how the sample size was determined. This needs to be included to enable readers to see if there is adequate power behind the study. No statistical analysis plans have been presented. Only one ethics approval reference has been included despite saying these samples come from three distinct studies. All previous studies should have there ethics approval reference included.

**Results**

-Does the analysis presented match the analysis plan?

-Are the results clearly and completely presented?

-Are the figures (Tables, Images) of sufficient quality for clarity?

Reviewer #1: (No Response)

Reviewer #2: My primary question relates to the broad time frame that the tested samples derived from. Since the positive samples were from before 2010, does that add confidence to your estimates of no/little yaws? Perhaps showing a distribution of the samples collected by year would be helpful. For example, are there no positive samples since 2010 because there are almost no samples from that time frame? Knowing that information will help us understand what the year of collection is really telling us about current yaws transmission.

Reviewer #3: • The table 1’s categories are unclear 

• You said that there were two false positive results, I would disagree these are false positive as they were presumably positive during repeat testing using the OnSite Syphilis Ab Combo Rapid Test. Why do you believe the DPP test to be more accurate? 

• The lateral flow tests are looking for antibodies but you mention results being positive for antigens. 

• “Seropositivity was observed in 0.4% of all children tested (active 0.1% and past infections 0.3%)” this is confusing, these percentage should not be added. 

• Statistics needed, you say “Rates of seropositivity using the confirmatory assay did not differ between formerly endemic and non-endemic samples of schoolchildren (0.4% - 0.1% active infections and 0.3% past infections)” but have not presented any statistics, it is also not clear what the seropositivity rate is in each community. 

• I think you need to be clear throughout the paper these are not active infections now, but at the time of the sero-survey.

• When you say “Most communities in the formerly endemic area had no evidence of T. pallidum seropositivity but 3 communities had evidence of past infections (seropositivity ranging 0.8-4.2%) of which 1 had evidence of an active infection” what do you mean by T.pallidum seropositivity, are you distinguishing this to a past infection? I think you need to be clearer in defining terms you go on to use throughout the paper. 

• Table 2 doesn’t add much that hasn’t already been described in the text

**Conclusions**

-Are the conclusions supported by the data presented?

-Are the limitations of analysis clearly described?

-Do the authors discuss how these data can be helpful to advance our understanding of the topic under study?

-Is public health relevance addressed?

Reviewer #1: (No Response)

Reviewer #2: Same as results - having a better understanding of the distribution of samples over the noted time frame will help us better understand the data in Table 2 - are no positive specimens since 2010 indicative of no transmission or were there not enough samples representing the time frame 2010-2017 to make any inference on recent yaws transmission.

Reviewer #3: You have concluded that the positive DPP tests found could be attributable to syphilis infection and that along with little evidence of clinical infections despite ongoing survelliance means yaws is likely to no longer be endemic. You have also mentioned how these data are important for declaring interrupted transmission and the future work that is needed to do this. Again the discussion needs proofing.

**Editorial and Data Presentation Modifications?**

Reviewer #1: (No Response)

Reviewer #2: (No Response)

Reviewer #3: This paper should be thoroughly proof read prior to re-submission as there are many grammatical errors and lots of very long sentences. It would benefit from the use of more scientific language in parts. Make sure you correctly use and are consistent with the terms: interruption of transmission, elimination and eradication. Include missing ethics references. Make sure terms are well defined.

**Summary and General Comments**

Reviewer #1: Major comments:

Methods

- Were the DPP tests read by eye or by the electronic reader?

- A map would be extremely helpful to provide a clearer understandign of the overlap between the districts and communities from which samples are available in the current study and the presumed formerly endemic districts in esmereldas. THe map provided I think (although not 100% clear from the figure legend) shows where the samples came from and if they were endemic or not endemic previously but doesnt (as far as I can tell) give a sense of what proportion of formerly endemic communities are represented in the current study.

Results 

Do I read the results correctly that of 53 tests positive on the initial treponemal anibody test (and which were positive on repeat testing) 33 were negative for the treponemal component of the DPP? This is quite unusual. 

Please provide at least some demographics - age, gender etc. 

Did the authors consider an age seroprevalence curve? It might be a useful supplementary figure. If there was active transmission of yaws one would anticipate a slow but consistent increase in treponemal positivity rates with age (as the test stays poisitve for life) whereas the pattern would look different in the context of venereal syphilis transmission. 

There should be confidence intervals on the proportions presented throughout the manuscript. 

Minor Comments:

Some typos - for example in the introduction "Frozen sera from these studies were selected based on living a previously endemic community for yaws" is missing a word. 

Im not quite sure what is meant by the sentence:

". Only latent yaws with serological evidence of active infection can sustain transmission" do you mean treponemal antibody positivity alone doesnt find potentially infectious individuals?

In the sentence "that have been shown to harbour T. pallidum pertenue-like spirochetes " Im not sure it is correct to use the word like. Phylogenetically the organism in NHPs IS t.p.pertenue.

Reviewer #2: (No Response)

Reviewer #3: This study is important for declaring elimination in Ecuador. However, there are current limitations including the lack of statistical analysis and the presentation of results as well as clear and consistent use of definitions such as seropositivity. There needs to be inclusion of a sample size calc, statistical analysis plan and presentation of these results.

PLOS authors have the option to publish the peer review history of their article (what does this mean?). If published, this will include your full peer review and any attached files.

Reviewer #1: No

Reviewer #2: No

Reviewer #3: Yes: Becca L. Handley

Figure Files:

Data Requirements:

Reproducibility:

References

---

## [Editor Report · Decision Letter 1]

19 Apr 2022

Dear Dr. Cooper,

We are pleased to inform you that your manuscript 'Yaws elimination in Ecuador: findings of a serological survey of children in Esmeraldas Province to evaluate interruption of transmission' has been provisionally accepted for publication in PLOS Neglected Tropical Diseases.

Best regards,

Brian Stevenson, Ph.D.

Associate Editor

Melissa Caimano

Deputy Editor

---

## [Editor Report · Acceptance letter]

7 May 2022

Dear Dr. Cooper,

We are delighted to inform you that your manuscript, "Yaws elimination in Ecuador: findings of a serological survey of children in Esmeraldas Province to evaluate interruption of transmission," has been formally accepted for publication in PLOS Neglected Tropical Diseases.

Best regards,

Shaden Kamhawi

co-Editor-in-Chief

Paul Brindley

co-Editor-in-Chief
